# Soft-Gated Warping-GAN for Pose-Guided Person Image Synthesis

**Haoye Dong**[1,2] , **Xiaodan Liang**[3,*] , **Ke Gong**[1] , **Hanjiang Lai**[1,2] , **Jia Zhu**[4] , **Jian Yin**[1,2]

[1]School of Data and Computer Science, Sun Yat-sen University
[2]Guangdong Key Laboratory of Big Data Analysis and Processing, Guangzhou 510006, P.R.China
[3]School of Intelligent Systems Engineering, Sun Yat-sen University
[4]School of Computer Science, South China Normal University
{donghy7@mail2, laihanj3@mail, issjyin@mail}.sysu.edu.cn
{xdliang328, kegong936}@gmail.com, jzhu@m.scun.edu.cn

## Abstract

Despite remarkable advances in image synthesis research, existing works often fail in manipulating images under the context of large geometric transformations. Synthesizing person images conditioned on arbitrary poses is one of the most representative examples where the generation quality largely relies on the capability of identifying and modeling arbitrary transformations on different body parts. Current generative models are often built on local convolutions and overlook the key challenges (e.g. heavy occlusions, different views or dramatic appearance changes) when distinct geometric changes happen for each part, caused by arbitrary pose manipulations. This paper aims to resolve these challenges induced by geometric variability and spatial displacements via a new Soft-Gated Warping Generative Adversarial Network (Warping-GAN), which is composed of two stages: 1) it first synthesizes a target part segmentation map given a target pose, which depicts the region-level spatial layouts for guiding image synthesis with higher-level structure constraints; 2) the Warping-GAN equipped with a soft-gated warping-block learns feature-level mapping to render textures from the original image into the generated segmentation map. Warping-GAN is capable of controlling different transformation degrees given distinct target poses. Moreover, the proposed warping-block is lightweight and flexible enough to be injected into any networks. Human perceptual studies and quantitative evaluations demonstrate the superiority of our Warping-GAN that significantly outperforms all existing methods on two large datasets.

## 1   Introduction

Person image synthesis, posed as one of most challenging tasks in image analysis, has huge potential applications for movie making, human-computer interaction, motion prediction, etc. Despite recent advances in image synthesis for low-level texture transformations [13, 35, 14] (e.g. style or colors), the person image synthesis is particularly under-explored and encounters with more challenges that cannot be resolved due to the technical limitations of existing models. The main difficulties that affect the generation quality lie in substantial appearance diversity and spatial layout transformations on clothes and body parts, induced by large geometric changes for arbitrary pose manipulations. Existing models [20, 21, 28, 8, 19] built on the encoder-decoder structure lack in considering the crucial shape and appearance misalignments, often leading to unsatisfying generated person images.

Among recent attempts of person image synthesis, the best-performing methods (PG2 [20], Body-ROI7 [21], and DSCF [28]) all directly used the conventional convolution-based generative models

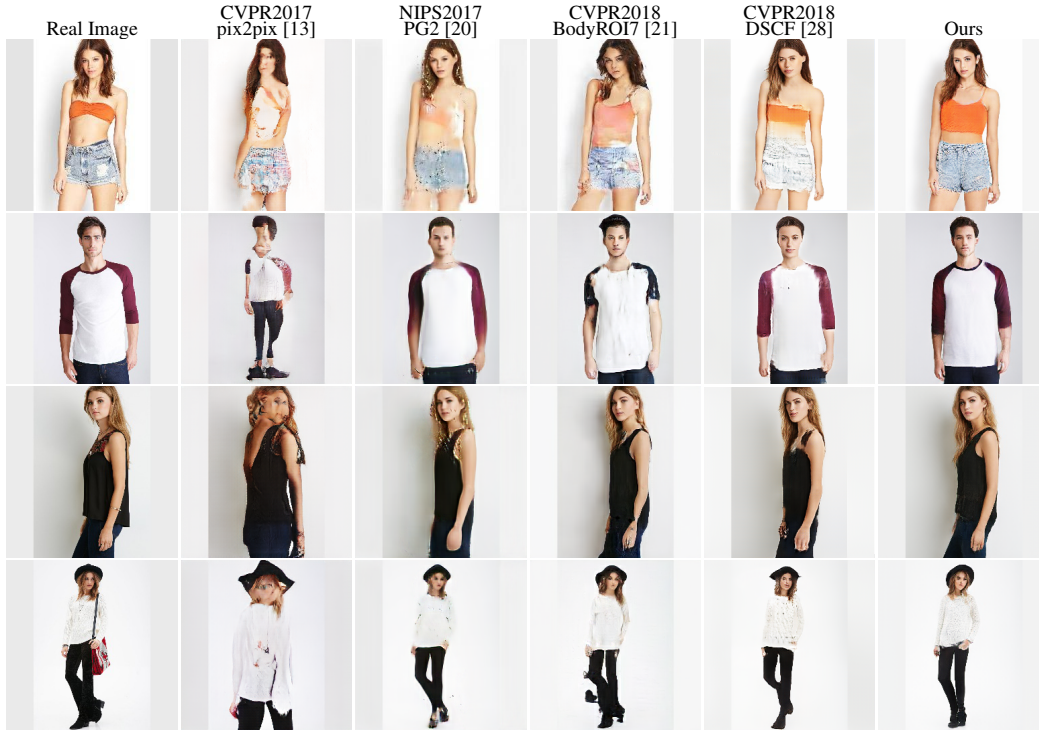

Figure 1: Comparison against the state-of-the-art methods on DeepFashion [36], based on the same condition images and the target poses. Our results are shown in the last column. Zoom in for details.

by taking either the image and target pose pairs or more body parts as inputs. DSCF [28] employed deformable skip connections to construct the generator and can only transform the images in a coarse rectangle scale using simple affinity property. However, they ignore the most critical issue (i.e. large spatial misalignment) in person image synthesis, which limits their capabilities in dealing with large pose changes. Besides, they fail to capture structure coherence between condition images with target poses due to the lack of modeling higher-level part-level structure layouts. Hence, their results suffer from various artifacts, blurry boundaries, missing clothing appearance when large geometric transformations are requested by the desirable poses, which are far from satisfaction. As the Figure 1 shows, the performance of existing state-of-the-art person image synthesis methods is disappointing due to the severe misalignment problem for reaching target poses.

In this paper, we propose a novel Soft-Gated Warping-GAN to address the large spatial misalignment issues induced by geometric transformations of desired poses, which includes two stages: 1) a pose-guided parser is employed to synthesize a part segmentation map given a target pose, which depicts part-level spatial layouts to better guide the image generation with high-level structure constraints; 2) a Warping-GAN renders detailed appearances into each segmentation part by learning geometric mappings from the original image to the target pose, conditioned on the predicted segmentation map. The Warping-GAN first trains a light-weight geometric matcher and then estimates its transformation parameters between the condition and synthesized segmentation maps. Based on the learned transformation parameters, the Warping-GAN incorporates a soft-gated warping-block which warps deep feature maps of the condition image to render the target segmentation map.

Our Warping-GAN has several technical merits. First, the warping-block can control the transformation degree via a soft gating function according to different pose manipulation requests. For example, a large transformation will be activated for significant pose changes while a small degree of transformation will be performed for the case that the original pose and target pose are similar. Second, warping informative feature maps rather than raw pixel values could help synthesize more realistic images, benefiting from the powerful feature extraction. Third, the warping-block can adaptively select effective feature maps by attention layers to perform warping.

Extensive experiments demonstrate that the proposed Soft-Gated Warping-GAN significantly outperforms the existing state-of-the-art methods on pose-based person image synthesis both qualitatively

and quantitatively, especially for large pose variation. Additionally, human perceptual study further indicates the superiority of our model that achieves remarkably higher scores compared to other methods with more realistic generated results.

## 2 Relation Works

**Image Synthesis**. Driven by remarkable results of GANs [10], lots of researchers leveraged GANs to generate images [12, 6, 18]. DCGANs [24] introduced an unsupervised learning method to effectively generate realistic images, which combined convolutional neural networks (CNNs) with GANs. Pix2pix [13] exploited a conditional adversarial networks (CGANs) [22] to tackle the image-to-image translation tasks, which learned the mapping from condition images to target images. CycleGAN [35], DiscoGAN [15], and DualGAN [33] each proposed an unsupervised method to generate the image from two domains with unlabeled images. Furthermore, StarGAN [5] proposed a unified model for image-to-image transformations task towards multiple domains, which is effective on young-to-old, angry-to-happy, and female-to-male. Pix2pixHD [30] used two different scales residual networks to generate the high-resolution images by two steps. These approaches are capable of learning to generate realistic images, but have limited scalability in handling posed-based person synthesis, because of the unseen target poses and the complex conditional appearances. Unlike those methods, we proposed a novel Soft-Gated Warping-GAN that pays attention to pose alignment in deep feature space and deals with textures rendering on the region-level for synthesizing person images.

**Person Image Synthesis**. Recently, lots of studies have been proposed to leverage adversarial learning for person image synthesis. PG2 [20] proposed a two-stage GANs architecture to synthesize the person images based on pose keypoints. BodyROI7 [21] applied disentangle and restructure methods to generate person images from different sampling features. DSCF [28] introduced a special U-Net [26] structure with deformable skip connections as a generator to synthesize person images from decomposed and deformable images. AUNET [8] presented a variational U-Net for generating images conditioned on a stickman (more artificial pose information), manipulating the appearance and shape by a variational Autoencoder. Skeleton-Aided [32] proposed a skeleton-aided method for video generation with a standard pix2pix [13] architecture, generating human images base on poses. [1] proposed a modular GANs, separating the image into different parts and reconstructing them by target pose. [23] essentially used CycleGAN [35] to generate person images, which applied conditioned bidirectional generators to reconstruct the original image by the pose. VITON [11] used a coarse-to-fine strategy to transfer a clothing image into a fixed pose person image. CP-VTON [29] learns a thin-plate spline transformation for transforming the in-shop clothes into fitting the body shape of the target person via a Geometric Matching Module (GMM). However, all methods above share a common problem, ignoring the deep feature maps misalignment between the condition and target images. In this paper, we exploit a Soft-Gated Warping-GAN, including a pose-guided parser to generate the target parsing, which guides to render textures on the specific part segmentation regions, and a novel warping-block to align the image features, which produces more realistic-look textures for synthesizing high-quality person images conditioned on different poses.

## 3 Soft-Gated Warping-GAN

Our goal is to change the pose of a given person image to another while keeping the texture details, leveraging the transformation mapping between the condition and target segmentation maps. We decompose this task into two stages: pose-guided parsing and Warping-GAN rendering. We first describe the overview of our Soft-Gated Warping-GAN architecture. Then, we discuss the pose-guided parsing and Warping-GAN rendering in details, respectively. Next, we present the warping-block design and the pipeline for estimating transformation parameters and warping images, which benefits to generate realistic-looking person images. Finally, we give a detailed description of the synthesis loss functions applied in our network.

### 3.1 Network Architectures

Our pipeline is a two-stage architecture for pose-guided parsing and Warping-GAN rendering respectively, which includes a human parsing parser, a pose estimator, and the affine [7]/TPS [2, 25] (Thin-Plate Spline) transformation estimator. Notably, we make the first attempt to estimate the

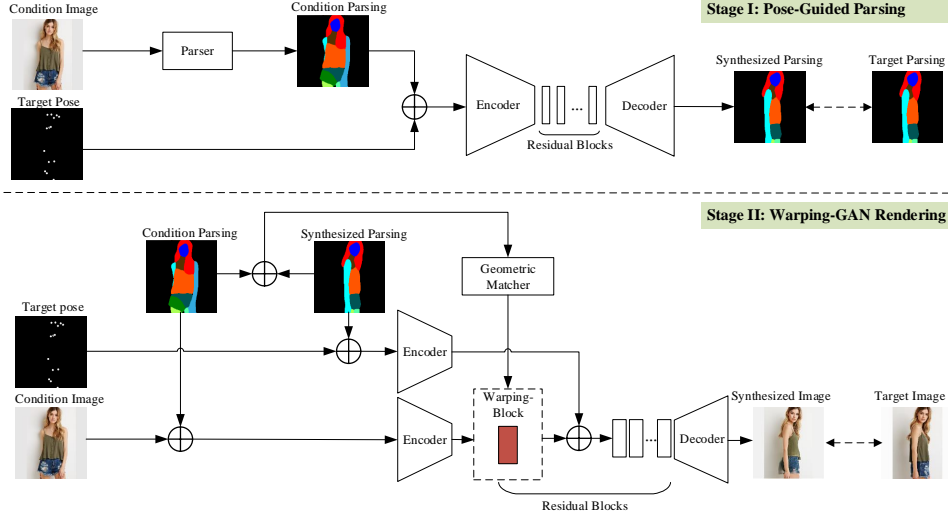

Figure 2: The overview of our Soft-Gated Warping-GAN. Given a condition image and a target pose, our model first generates the target parsing using a pose-guided parser. We then estimate the transformations between the condition and target parsing by a geometric matcher following a soft-gated warping-block to warp the image features. Subsequently, we concatenate the warped feature maps, embedded pose, and synthesized parsing to generate the realistic-looking image.

transformation of person part segmentation maps for generating person images. In stage I, we first predict the human parsing based on the target pose and the parsing result from the condition image. The synthesized parsing result is severed as the spatial constraint to enhance the person coherence. In stage II, we use the synthesized parsing result from the stage I, the condition image, and the target pose jointly to trained a deep warping-block based generator and a discriminator, which is able to render the texture details on the specific regions. In both stages, we only take the condition image and the target pose as input. In contrast to AUNET [8](using 'stickman' to represent pose, involving more artificial and constraint for training), we are following PG2 [20] to encode the pose with 18 heatmaps. Each heatmap has one point that is filled with 1 in 4-pixel radius circle and 0 elsewhere.

### 3.1.1 Stage I: Pose-Guided Parsing

To learn the mapping from condition image to the target pose on a part-level, a pose-guide parser is introduced to generate the human parsing of target image conditioned on the pose. The synthesized human parsing contains pixel-wise class labels that can guide the image generation on the class-level, as it can help to refine the detailed appearance of parts, such as face, clothes, and hands. Since the DeepFashion and Market-1501 dataset do not have human parsing labels, we use the LIP [9] dataset to train a human parsing network. The LIP [9] dataset consists of 50,462 images and each person has 20 semantic labels. To capture the refined appearance of the person, we transfer the synthesized parsing labels into an one-hot tensor with 20 channels. Each channel is a binary mask vector and denotes the one class of person parts. These vectors are trained jointly with condition image and pose to capture the information from both the image features and the structure of the person, which benefits to synthesize more realistic-looking person images. Adapted from Pix2pix [13], the generator of the pose-guided parser contains 9 residual blocks. In addition, we utilize a pixel-wise softmax loss from LIP [9] to enhance the quality of results. As shown in Figure 2, the pose-guided parser consists of one ResNet-like generator, which takes condition image and target pose as input, and outputs the target parsing which obeys the target pose.

### 3.1.2 Stage II: Warping-GAN Rendering

In this stage, we exploit a novel region-wise learning to render the texture details based on specific regions, guided by the synthesized parsing from the stage I. Formally, Let $I_i = P(l_i)$ denote the function for the region-wise learning, where $I_i$ and $l_i$ denote the $i$-th pixel value and the class label of this pixel respectively. And $i$ ($0 \leq i < n$) denotes the index of pixel in image. $n$ is the total number

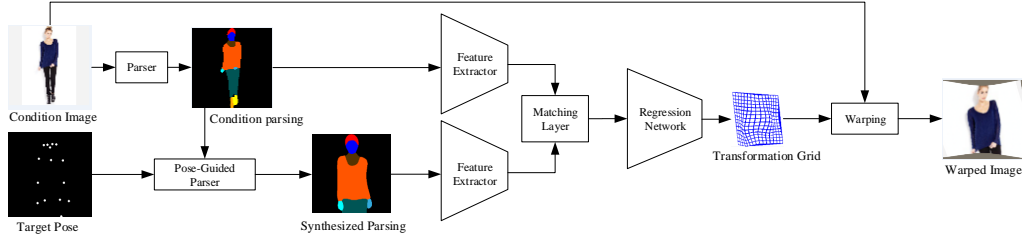

Figure 3: The architecture of Geometric Matcher. We first produce a condition parsing from the condition image by a parser. Then, we synthesize the target parsing with the target pose and the condition parsing. The condition and synthesized parsing are passed through two feature extractors, respectively, following a feature matching layer and a regression network. The condition image is warped using the transformation grid in the end.

of pixels in one image. Note that in this work, the segmentation map also named parsing or human parsing, since our method towards the person images.

However, the misalignments between the condition and target image lead to generate blurry values. To alleviate this problem, we further learn the mapping between condition image and target pose by introducing two novel approaches: the geometric matcher and the soft-gated warping-block transfer. Inspired by the geometric matching method, GEO [25], we propose a parsing-based geometric matching method to estimate the transformation between the condition and synthesized parsing. Besides, we design a novel block named warping-block for warping the condition image on a part-level, using the synthesized parsing from stage I. Note that, those transformation mappings are estimated from the parsing, which we can use to warp the deep features of condition image.

**Geometric Matcher.** We train a geometric matcher to estimate the transformation mapping between the condition and synthesized parsing, as illustrate in Figure 3. Different from GEO [25], we handle this issue as parsing context matching, which can also estimate the transformation effectively. Due to the lack of the target image in the test phrase, we use the condition and synthesized parsing to compute the transformation parameters. In our method, we combine affine and TPS to obtain the transformation mapping through a siamesed convolutional neural network following GEO [25]. To be specific, we first estimate the affine transformation between the condition and synthesized parsing. Based on the results from affine estimation, we then estimate TPS transformation parameters between warping results from the affine transformation and target parsing. The transformation mappings are adopted to transform the extracted features of the condition image, which helps to alleviate the misalignment problem.

**Soft-gated Warping-Block.** Inspired by [3], having obtained the transformation mapping from the geometric matcher, we subsequently use this mapping to warp deep feature maps, which is able to capture the significant high-level information and thus help to synthesize image in an approximated shape with more realistic-looking details. We combine the affine [7] and TPS [2] (Thin-Plate Spline transformation) as the transformation operation of the warping-block. As shown in Figure 4, we denote those transformations as the transformation grid. Formally, let $\Phi(I)$ denotes the deep feature map, $R(\Phi(I))$ denotes

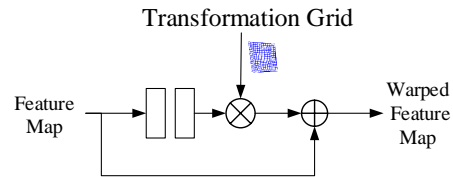

Figure 4: The architecture of soft-gated warping-block. Zoom in for details.

the residual feature map from $\Phi(I)$, $W(I)$ represents the operation of the transformation grid. Thus, we regard $T(I)$ as the transformation operation, we then formulate the transformation mapping of the warping-block as:

$$T(\Phi(I)) = \Phi(I) + W(I) \cdot R(\Phi(I)), \tag{1}$$

where $\cdot$ denotes the matrix multiplication, we denote $e$ as the element of $W(I)$, $e \in [0, 1]$, which acts as the soft gate of residual feature maps to address the misalignment problem caused by different poses. Hence, the warping-block can control the transformation degree via a soft-gated function

according to different pose manipulation requests, which is light-weight and flexible to be injected into any generative networks.

### 3.1.3 Generator and Discriminator

**Generator.** Adapted from pix2pix [13], we build two residual-like generators. One generator in stage I contains standard residual blocks in the middle, and another generator adds the warping-block following encoder in stage II. As shown in Figure 2, Both generators consist of an encoder, a decoder and 9 residual blocks.

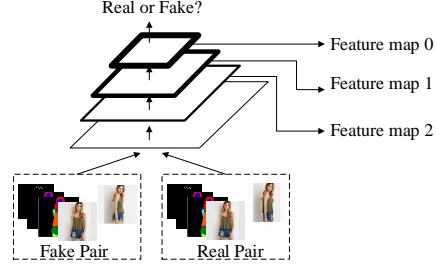

Figure 5: The overview of discriminator in stage II. Zoom in for details.

**Discriminator.** To achieve a stabilized training, inspired by [30], we adopt the pyramidal hierarchy layers to build the discriminator in both stages, as illustrated in Figure 5. We combine condition image, target keypoints, condition parsing, real/synthesized parsing, and real/synthesized image as input for the discriminator. We observe that feature maps from the pyramidal hierarchy discriminator benefits to enhance the quality of synthesized images. More details are shown in the following section.

## 3.2 Objective Functions

We aim to build a generator that synthesizes person image in arbitrary poses. Due to the complex details of the image and the variety of poses, it's challenging for training generator well. To address these issues, we apply four losses to alleviate them in different aspects, which are adversarial loss $\mathcal{L}_{\text{adv}}$ [10], pixel-wise loss $\mathcal{L}_{\text{pixel}}$ [32], perceptual loss $\mathcal{L}_{\text{perceptual}}$ [14, 11, 17] and pyramidal hierarchy loss $\mathcal{L}_{\text{PH}}$. As the Fig 5 shown, the pyramidal hierarchy contains useful and effective feature maps at different scales in different layers of the discriminator. To fully leverage these feature maps, we define a pyramidal hierarchy loss, as illustrated in Eq. 2.

$$\mathcal{L}_{\text{PH}} = \sum_{i=0}^{n} \alpha_i \|F_i(\hat{I}) - F_i(I)\|_1, \tag{2}$$

where $F_i(I)$ denotes the $i$-th $(i = 0, 1, 2)$ layer feature map from the trained discriminator. We also use L1 norm to calculate the losses of feature maps of each layer and sum them with the weight $\alpha_i$. The generator objective is a weighted sum of different losses, written as follows.

$$\mathcal{L}_{\text{total}} = \lambda_1 \mathcal{L}_{\text{adv}} + \lambda_2 \mathcal{L}_{\text{pixel}} + \lambda_3 \mathcal{L}_{\text{perceptual}} + \lambda_4 \mathcal{L}_{\text{PH}}, \tag{3}$$

where $\lambda_i$ denotes the weight of $i$-loss, respectively.

## 4 Experiments

We perform extensive experiments to evaluate the capability of our approach against recent methods on two famous datasets. Moreover, we further perform a human perceptual study on the Amazon Mechanical Turk (AMT) to evaluate the visualized results of our method. Finally, we demonstrate an ablation study to verify the influence of each important component in our framework.

### 4.1 Datasets and Implementation Details

**DeepFashion** [36] consists of 52,712 person images in fashion clothes with image size $256\times256$. Following [20, 21, 28], we remove the failure case images with pose estimator [4] and human parser [9], then extract the image pairs that contain the same person in same clothes with two different poses. We select 81,414 pairs as our training set and randomly select 12,800 pairs for testing.

**Market-1501** [34] contains 322,668 images collected from 1,501 persons with image size $128\times64$. According to [34, 20, 21, 28], we extract the image pairs that reach about 600,000. Then we also randomly select 12,800 pairs as the test set and 296,938 pairs for training.

Table 1: Comparison on DeepFashion and Market-1501 datasets.

| Model | DeepFashion [36] | | Market-1501 [34] | |
|---|---|---|---|---|
| | SSIM | IS | SSIM | IS |
| pix2pix [13] (CVPR2017) | 0.692 | 3.249 | 0.183 | 2.678 |
| PG2 [20] (NIPS2017) | 0.762 | 3.090 | 0.253 | 3.460 |
| DSCF [28] (CVPR2018) | 0.761 | 3.351 | 0.290 | 3.185 |
| UPIS [23] (CVPR2018) | 0.747 | 2.97 | – | – |
| AUNET [8] (CVPR2018) | 0.786 | 3.087 | 0.353 | 3.214 |
| BodyROI7 [21] (CVPR2018) | 0.614 | 3.228 | 0.099 | **3.483** |
| w/o parsing | 0.692 | 3.146 | 0.236 | 2.489 |
| w/o soft-gated warping-block | 0.777 | 3.262 | 0.337 | 3.394 |
| w/o $\mathcal{L}_{adv}$ | 0.780 | 3.430 | 0.346 | 3.332 |
| w/o $\mathcal{L}_{perceptual}$ | 0.772 | **3.446** | 0.319 | 3.407 |
| w/o $\mathcal{L}_{pixel}$ | 0.780 | 3.270 | 0.337 | 3.292 |
| w/o $\mathcal{L}_{PH}$ | 0.776 | 3.323 | 0.337 | 3.448 |
| Ours (full) | **0.793** | 3.314 | **0.356** | 3.409 |

Table 2: Pairwise comparison with other approaches. Chance is at 50%. Each cell lists the percentage where our result is preferred over the other method.

| | pix2pix [13] | PG2 [20] | BodyROI7 [21] | DSCF [28] |
|---|---|---|---|---|
| DeepFashion [36] | 98.7% | 87.3% | 96.3% | 79.6% |
| Market-1501 [34] | 83.4% | 67.7% | 68.4% | 62.1% |

**Evaluation Metrics**: We use the Amazon Mechanical Turk (AMT) to evaluate the visual quality of synthesized results. We also apply Structural SIMilarity (SSIM) [31] and Inception Score (IS) [27] for quantitative analysis.

**Implementation Details**. Our network architecture is adapted from pix2pixHD [30], which has presented remarkable results for synthesizing images. The architecture consists of an generator with encoder-decoder structure and a multi-layer discriminator. The generator contains three downsampling layers, three upsampling layers, night residual blocks, and one Warping-Block block. We apply three different scale convolutional layers for the discriminator, and extract features from those layers to compute the Pyramidal Hierarchy loss (PH loss), as the Figure 5 shows. We use the Adam [16] optimizer and set $\beta_1 = 0.5$, $\beta_2 = 0.999$. We use learning rate of 0.0002. We use batch size of 12 for Deepfashion, and 20 for Market-1501.

## 4.2 Quantitative Results

To verify the effectiveness of our method, we conduct experiments on two benchmarks and compare against six recent related works. To obtain fair comparisons, we directly use the results provided by the authors. The comparison results and the advantages of our approach are also clearly shown in the numerical scores in Table 1. Our proposed method consistently outperforms all baselines for the SSIM metric on both datasets, thanks to the Soft-Gated Warping-GAN which can render high-level structure textures and control different transformation for the geometric variability. Our network also achieves comparable results for the IS metric on two datasets, which confirms the generalization ability of our Soft-Gated Warping-GAN.

## 4.3 Human Perceptual Study

We further evaluate our algorithm via a human subjective study. We perform pairwise A/B tests deployed on the Amazon Mechanical Turk (MTurk) platform on the DeepFashion [36] and the Market-1501 [34]. The workers are given two generated person images at once. One is synthesized by our method and the other is produced by the compared approach. They are then given unlimited time to select which image looks more natural. Our method achieves significantly better human

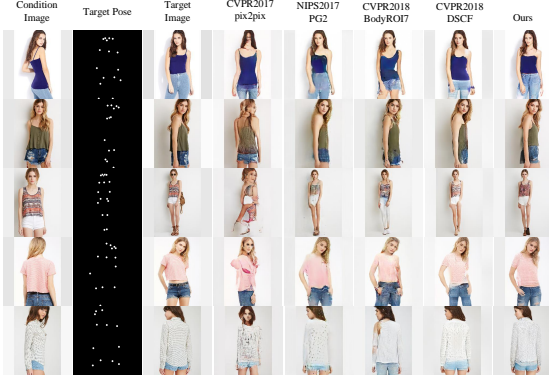 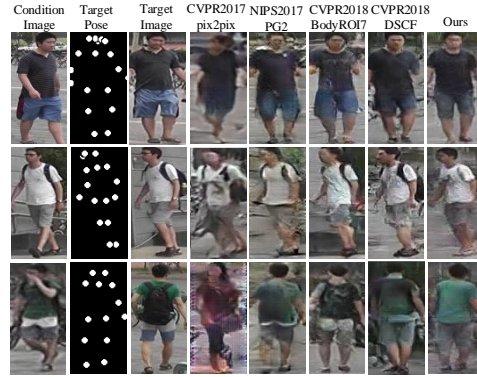

Figure 6: Comparison against the recent state-of-the-art methods, based on the same condition image and the target pose. Our results on DeepFashion [36] are shown in the last column. Zoom in for details.

Figure 7: Comparison against the recent state-of-the-art methods, based on the same condition image and the target pose. Our results on Market-1501 [34] shown in the last column.

evaluation scores, as summarized in Table 2. For example, compared to BodyROI7 [21], 96.3% workers determine that our method generated more realistic person images on DeepFashion [36] dataset. This superior performance confirms the effectiveness of our network comprised of a human parser and the soft-gated warping-block, which synthesizes more realistic and natural person images.

## 4.4 Qualitative Results

We next present and discuss a series of qualitative results that will highlight the main characteristics of the proposed approach, including its ability to render textures with high-level semantic part segmentation details and to control the transformation degrees conditioned on various poses. The qualitative results on the DeepFashion [36] and the Market-1501 [34] are visualized in Figure 6 and Figure 7. Existed methods create blurry and coarse results without considering to render the details of target clothing items by human parsing. Some methods produce sharper edges, but also cause undesirable artifacts missing some parts. In contrast to these methods, our approach accurately and seamlessly generates more detailed and precise virtual person images conditioned on different target poses. However, there are some blurry results on Market-1501 [34], which might result from that the performance of the human parser in our model may be influenced by the low-resolution images. We will present and analyze more visualized results and failure cases in the supplementary materials.

## 4.5 Ablation study

To verify the impact of each component of the proposed method, we conduct an ablation study on DeepFashion [36] and Market-1501 [34]. As shown in Table 1 and Figure 8, we report evaluation results of the different versions of the proposed method. We first compare the results using pose-guided parsing to the results without using it. From the comparison, we can learn that incorporating the human parser into our generator significantly improves the performance of generation, which can depict the region-level spatial layouts for guiding image synthesis with higher-level structure constraints by part segmentation maps. We then examine the effectiveness of the proposed soft-gated warping-block. From Table 1 and Figure 8, we observe that the performance drops dramatically without the soft-gated warping-block. The results suggest the improved performance attained by the warping-block insertion is not merely due to the additional parameters, but the effective mechanism inherently brought by the warping operations which act as a soft gate to control different transformation degrees according to the pose manipulations. We also study the importance of each term in our objective function. As can be seen, adding each of the four losses can substantially enhance the results.

## 5 Conclusion

In this work, we presented a novel Soft-Gated Warping-GAN for addressing pose-guided person image synthesis, which aims to resolve the challenges induced by geometric variability and spatial

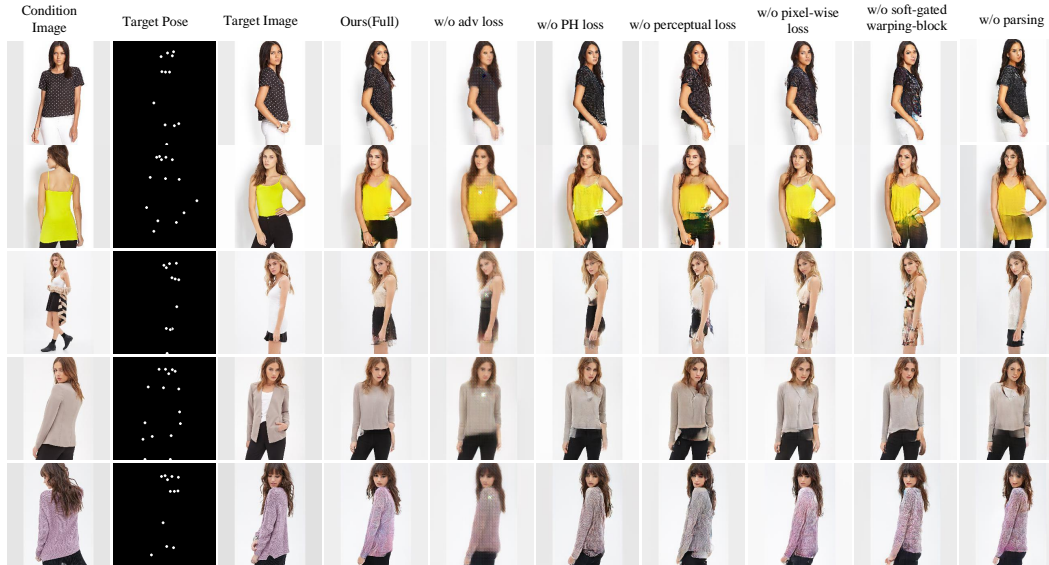

| Condition Image | Target Pose | Target Image | Ours(Full) | w/o adv loss | w/o PH loss | w/o perceptual loss | w/o pixel-wise loss | w/o soft-gated warping-block | w/o parsing |

Figure 8: Ablation studies on DeepFashion [36]. Zoom in for details.

displacements. Our approach incorporates a human parser to produce a target part segmentation map to instruct image synthesis with higher-level structure information, and a soft-gated warping-block to warp the feature maps for rendering the textures. Effectively controlling different transformation degrees conditioned on various target poses, our proposed Soft-Gated Warping-GAN can generate remarkably realistic and natural results with the best human perceptual score. Qualitative and quantitative experimental results demonstrate the superiority of our proposed method, which achieves state-of-the-art performance on two large datasets.

## Acknowledgements

This work is supported by the National Natural Science Foundation of China (61472453, U1401256, U1501252, U1611264, U1711261, U1711262, 61602530), and National Natural Science Foundation of China (NSFC) under Grant No. 61836012.

## Footnotes

*Corresponding author is Xiaodan Liang

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
