[Supplementary Material]

# Supplementary Material: Soft-Gated Warping-GAN for Pose-Guided Person Image Synthesis

Figure 1: Some results of different target poses with the same condition image (first column) on DeepFashion [5].

Figure 2: Compare against with BodyROI7 [2] (CVPR2018) on DeepFashion [5].

Figure 3: Compare against with DSCF [3] (CVPR2018) on DeepFashion [5]. Zoom in for details.

Figure 4: Compare against with PG2 [1] (NIPS2017) on DeepFashion [5]. Zoom in for details.

| Condition Image | Target Pose | Target Image | Ours | CVPR2018 BodyROI7 | Condition Image | Target Pose | Target Image | Ours | CVPR2018 BodyROI7 |
|---|---|---|---|---|---|---|---|---|---|

Figure 5: Compare against with BodyROI7 [2] (CVPR2018) on Market-1501 [4]. Zoom in for details.

| Condition Image | Target Pose | Target Image | Ours | CVPR2018 DSCF | Condition Image | Target Pose | Target Image | Ours | CVPR2018 DSCF |
|---|---|---|---|---|---|---|---|---|---|

Figure 6: Compare against with DSCF [3] (CVPR2018) on Market-1501 [4]. Zoom in for details.

Figure 7: Compare against with PG2 [1] (NIPS2017) on Market-1501 [4]. Zoom in for details.