[Reviews · NeurIPS 2018]

Reviewer 1



This paper presents a parsing-guided deep network for re-posing a person image. In particular, the network takes as inputs a person image and a novel pose and then produces a novel image of the same person in the novel pose. The proposed network consists of two stages. In the first stage, a off-the-shelf human parser is used to segment the input image into parts. Given the part segmentation and the target pose, a pix2pix network is trained to synthesize target part segmentation masks or parsing. In the second stage, the source and target part masks are used to find affine and TPS transformations so that the appearance features of the input image are warped to align with the target pose to synthesize the final output. Results are reported on a fashion image dataset and a surveillance dataset where most of images have the stand-up poses. It also compares a number of recent methods and evaluates results using SSIM, Inception scores and AMT. +It is an interesting idea to use part segmentation masks to align the input image with the target pose. In order to do that, this paper first trained a target mask synthesis network which could make it easy to synthesize the final image. In terms of using part segmentation for human image re-posing, this paper is closely related to [1]. But there are several differences. First, this paper directly synthesize the target segmentation in a supervised network while [1] learns the part segmentation in an unsupervised manner. Second, [1] uses affine transformation while this paper also uses TPS transformation. Third, this paper warps features while [1] warps pixels directly. +Results on DeepFashion dataset are impressive. There are much less distortions then previous methods. Evaluations using SSIM, IS and AMT again a number of previous methods also agree with the visual results. -As said above, this paper is closely related to [1]. It should discuss the differences clearly. It will be great if this paper can compare with [1] in experiments, which will significantly strengthen the contributions. TPS transformation is a novel component. I believe it is helpful for cloth warping. But unfortunately, there is no ablation study for this (with or without TPS). -The deformable GANs by S. Siarohin et al. in CVPR2018 should be also discussed as it applies part affine transformations to feature maps. -To make the paper self-contained, it is better to introduce the core algorithm of GEO briefly. -Pix2pix[10] may not be a good baseline as it is not designed for re-posing. Instead, the analogy-making network used in [Ruben Villegas et al. ICML2017] could serve a better baseline. -It seems like the input target pose has to have the same skeleton structure as the input person (in terms bone length and proportions). However, obtaining such input pose itself is non-trivial. Directly manipulating the input pose may result in 3D impossible poses. Extracting the pose from another person image will expose a retargeting problem. This could be a limitation that other papers also have. It will be good to discuss it in the paper. -Some sentences are fragmented. “Applied disentangle and restructure”, “large arrange among” -It is not clear to me why the proposed method is termed as "soft-gated". This term is only referred in a headline in the main algorithm section, but never formally discussed. -Both DeepFashion and Market-1501 have constrained poses. It will be interesting to test on other datasets with more pose variations such as PennAction.

Reviewer 2



The authors propose a computer vision framework for performing pose transfer, i.e. generating images of a given person in a random pose. The key technical contributions of this approach are 1) exploiting an external human parser for an intermediate step of predicting target segmentations given the input and the target pose, 2) a warping module estimating transformations between the input and the predicted target human parsings. In my opinion, the paper is rather interesting and original: I like the idea of explicitly learned transformations and its adaptation for the task. There have been a number of works around modeling deformations (for example, [23] estimated local affined transformations for body parts), but the proposed implementation went further by learning transformation fields densely and end-to-end. The experimental results show significant improvements over the state-of-the-art given the chosen standard benchmarks and the metrics. The manuscript is rather easy to follow and understand, but it still contains a significant number of typos and minor grammar mistakes. Some implementation details are missing which may make this work difficult to reproduce. It would be helpful if the authors provided a more detailed description of the geometric matcher. My high level concern, however, is that there has been a large number of similar works optimizing for performance (overfitting?) on DeepFashion/Market-1501 datasets, which have rather low variety and cannot be considered representative of real data. I personally would be excited to see us as a community move beyond these benchmarks and target more ambitious settings. My more detailed comments and questions are below. The human parser has been trained on LIP dataset constructed from COCO images - therefore, I'd like to see some comments on the quality of obtained segmentations (on DeepFashion, and especially on the low resolution Market-1501 dataset) and how it affects the quality of the generations. Stage 1 & Figure 2. Does the input to the trained parser include the original RGB image? L130-131 suggest so, while Figure 2 doesn't. Are heatmaps and probability maps concatenated? How is the background label treated? Figure 3. Is the Geometric Matcher module trained separately or jointly with the rest of the generator? What is the meaning of the block on the right of the Matching layer? Adding labels to this figure would be helpful. Geometric Matcher & Warping Block. The abbreviation TPS is used before it's defined. What makes the network generalize on the body parts that are not observed in the source image (for example, going from front to back)? The description of the warping block would be easier to understand if the authors showed the corresponding notations in Figure 4. Why is learning facilitated by decomposing the result into the original set of features and residuals? How is the network's performance affected if the warping is done directly? The term "soft gate" is unclear in this context and its position in Figure 4 is not obvious. Discriminator architecture. In the proposed implementation, the discriminator takes an exhaustive set of inputs (input image, target keypoints, source parsing, target parsing, generated image) - have the authors analyzed the importance of each of these components? Objective functions. It would be easier for the reader if the authors explicitly described their loss terms (L1 or L2 for L_pixel, base network for L_perceptual, the exact kind of GAN objective). Stabilizing discriminators by matching intermediate features at multiple scales has been done before - the authors may want to cite: Wang et al. High-Resolution Image Synthesis and Semantic Manipulation with Conditional GANs. CVPR, 2018. Datasets. How is the parser used to remove failure cases from the dataset, assuming that there is no ground truth for this data? How is the training / test splits are defined? The number of training / test images in this paper differs from, for example, [23]. Assuming it was arbitrary, did the authors retrain the state-of-the-art models on the same split? Otherwise, the comparison may be unfair (images from the training set used for evaluation). Symbol beta is overloaded (used in Eq. (2) and also to define Adam's parameters). How are betas in (2) are defined? Using Inception score has been a common place in similar works, but one should keep in mind that it's not strictly appropriate for the task and in practice is highly unstable (can be high for "blurry" images obtained with L1/L2 loss). I'm not sure what the authors mean saying that high IS scores confirm generalization (L239). Also, it would be useful to report IS for the real images. Human study: what was the exact protocol of the experiments? (number of images, number of workers, number of experiments / worker, how the images were selected, image resolution). It is somewhat disappointing that in these experiments the workers were asked to evaluate the "realism" of the generations, but not how well the target pose is captured or whether the identity and the appearance of the person from the original image are preserved. It would be interesting to see more results on these aspects as well. Why doesn't Table 2 contain comparison of the proposed method's outputs with ground truth images? Nitpicking: L119: we ARE following L136: please reformulate "parsing obey the input pose" L142: number of pixelS L162: Obtained -> Having obtained? L197: please rephrase "large arrange among of poses" L228: Figure shown -> Figure shows L246: significantLY bettter L268: generator significantly improveS Paper formatting seems to have suffered by reducing the manuscript to 8 pages, please fix.

Reviewer 3



Quality This manuscript proposes a soft-gated Warping-GAN for image translation conditioned on human pose. The approach has two stages, first use encoder-decoder to reconstruct target human parsing, second use GEO to estimation affine and tps transformation between source and target human parsing map, and then apply transformation on source feature map. The method is plausible, and results quite good compared with recent papers. The experimenting is quite sufficient. My major concern is about the method novelty, the idea that apply two stages or multiple stages and utilize human parsing is not completely new (e.g., VITON); the idea of Warping-GAN is also not completely new (e.g., “LR-GAN: Layered Recursive Generative Adversarial Networks for Image Generation”, “Deformable GAN”, “Semi-parametric Image Synthesis”, etc all learn or estimate warping parameters in encoder-decoder setting); the idea of pose-guided parsing is already implemented in LIP paper in a sense. All these individual modules seem to be utilized in various papers, but the combination, to my knowledge, is new. Minor questions: 1. If the two human pose has totally different view, like Fig 3, for example, one with full body and the other with lower body, how the GEO matcher will work? Have you considered any body part level transformation (like Deformable GAN)? 2. Is there comparison with Variational U-Net for this task? Clarity The paper is well-written and easy to understand, the experiments and ablation study is good. Significance In general, this manuscript has merit and contribution to the community, the paper is complete and experiments convincing, the novelty is somehow incremental but I think it’s still a good submission.